# Development of an Open-Source Thermal Image Processing Software for Improving Irrigation Management in Potato Crops (*Solanum tuberosum* L.)

**DOI:** 10.3390/s20020472

**Published:** 2020-01-14

**Authors:** Gonzalo Cucho-Padin, Javier Rinza, Johan Ninanya, Hildo Loayza, Roberto Quiroz, David A. Ramírez

**Affiliations:** 1Department of Electrical and Computer Engineering, University of Illinois at Urbana-Champaign, Champaign, IL 61801, USA; 2International Potato Center, Apartado 1558, Lima 12, Peru; j.rinza@cgiar.org (J.R.); j.ninanya@cgiar.org (J.N.); h.loayza@cgiar.org (H.L.); 3CATIE-Tropical Agricultural Research and Higher Education Center, Cartago Turrialba 30501, Costa Rica; roberto.quiroz@catie.ac.cr

**Keywords:** infrared thermography, image processing, canopy temperature, crop water stress index

## Abstract

Accurate determination of plant water status is mandatory to optimize irrigation scheduling and thus maximize yield. Infrared thermography (IRT) can be used as a proxy for detecting stomatal closure as a measure of plant water stress. In this study, an open-source software (Thermal Image Processor (TIPCIP)) that includes image processing techniques such as thermal-visible image segmentation and morphological operations was developed to estimate the crop water stress index (CWSI) in potato crops. Results were compared to the CWSI derived from thermocouples where a high correlation was found ($r_{Pearson}$ = 0.84). To evaluate the effectiveness of the software, two experiments were implemented. TIPCIP-based canopy temperature was used to estimate CWSI throughout the growing season, in a humid environment. Two treatments with different irrigation timings were established based on CWSI thresholds: 0.4 (T2) and 0.7 (T3), and compared against a control (T1, irrigated when soil moisture achieved 70% of field capacity). As a result, T2 showed no significant reduction in fresh tuber yield (34.5 ± 3.72 and 44.3 ± 2.66 t ha^−1^), allowing a total water saving of 341.6 ± 63.65 and 515.7 ± 37.73 m^3^ ha^−1^ in the first and second experiment, respectively. The findings have encouraged the initiation of experiments to automate the use of the CWSI for precision irrigation using either UAVs in large settings or by adapting TIPCIP to process data from smartphone-based IRT sensors for applications in smallholder settings.

## 1. Introduction

The relationship between water contents in the plant and yield has been well established [1,2]. The use of standardized parameters based on physiological thresholds that lead to photosynthetic impairment if surpassed have been successfully used [3,4,5,6]. Nevertheless, assessing the water status of a few leaves seldom represents the water status of the field, and thus irrigation decisions would benefit from noninvasive and nondestructive methods covering larger ranges [7,8]. The use of remote sensing for assessing plant water status on different scales, providing near real-time assessments, might become an adequate option. Notwithstanding, the approach has to be tested under field conditions with crops sensitive to water deficiency. Potato (*Solanum tuberosum* L.)—considered to be the third most prevalent edible crop in production after wheat and rice [9]—was used as an example. Its sensitivity to water scarcity (highly associated with a shallow root system) [10] and its cultivation in drought-prone zones caused by climate change [11] are the drivers spurring research to find ways to maximize potato yields while reducing water resources. The use of physiological thresholds has contributed to this goal; thus, Ramírez et al. [6] recommend keeping leaves at maximum stomatal conductance (associated with leaf pore openness and gas exchange) at saturated light >0.15 mol H_2_O m^−2^ s^−1^ to guarantee appropriate tuber yield whilst saving water. However, the use of this kind of physiological traits for water status inspection requires expensive, exhaustive, invasive, yet microscale (leaf) assessments. Thus, infrared thermography is seen as a promising technology that can detect variations in the canopy temperature ($T_{canopy}$), and ultimately, estimate the crop water stress index (CWSI), an indicator highly correlated with stomatal conductance [6,8,12] (see Section 2.4 for further details).

On the basis of the black body radiation theory, an infrared thermography (IRT) camera can detect the spectral radiance emitted by an object due to its temperature [13,14,15]. Such a radiance, which ranges from 7 to 14 μm, is converted into electrical signals and then displayed as a 2D array or IRT image. One of the shortcoming of current IRT image sensors is their typical resolution of 320 × 240 pixels, which precludes appropriate identification of the spatial structure in the captured scene. Unlike its CCD -based visible RGB counterpart, IRT sensors are based on a microbolometer, a sophisticated array of small thermal sensors with a complicated manufacturing process [16]. To overcome this problem, companies recently started to offer IRT camera systems that include a fixed visual (RGB) camera to help identify regions of interest through additional image processing as well as to avoid convoluted image registration methods [17,18].

Numerous agricultural studies have reported the use of infrared thermography along with particular acquisition methodologies to analyze different plant parameters and conditions. For example, Pitarma et al. [19] evaluated tree health through the use of IRT images. In this study, identification of healthy and unhealthy tissues in the tree’s surface was performed through an analysis of infrared thermographic data. Similarly, in [20], a review of pest detection applications based on infrared thermography was reported. This comprehensive study includes the detection and identification of the main pest in plant crops such as maize, rice, and soybean. With regard to water-stress estimation in plants, Möller et al. [21] conducted an investigation on the water status of grapevines using thermal and visible images acquired from a platform located 15 m above the canopy (nadir-pointing view). This study aimed to analyze the correlation between the calculated CWSI and the measured leaf stomatal conductance, resulting in a correlation of $R^{2}=0.91$. Image processing steps in the estimation of $T_{canopy}$ include image registration (i.e., alignment of multisensor images), which requires the placement of aluminum crosses over the field, and the selection of canopy pixels in the thermal image based on the transformation of the RGB image to hue-saturation-intensity color space followed by a manual thresholding procedure. A similar study was conducted in [22] for cotton crops where leaf water potentials showed a linear relationship with CWSI values ($R^{2}=0.816$). Here, a nadir-pointing view IRT camera was located 5 m above the crops. Since there was no additional sensor to detect where the canopy was located in the scene, canopy pixels detection was based solely on estimated temperature thresholds, thus avoiding the use of an additional sensor. In the context of potato crops, Prashar et al. [23] implemented a high-throughput field phenotyping methodology to estimate CWSI using IRT images. Image acquisition was performed from a fork-lift at a height of about 8 m, with an angular inclination to cover up to 27–36 plots in the scene. Then, manual selection of the central regions in the plots was performed to estimate the CWSI. Similarly to the previous study, the inclusion of canopy pixels in the analysis was based on temperature thresholding.

Consequently, several factors can influence the accurate estimation of $T_{canopy}$ and the subsequent calculation of the CWSI. First, the use of an independent, nonfixed sensor imposes an image registration problem where convoluted image processing algorithms could be required. The use of control points in the scene could alleviate the computational load while increasing the overall acquisition time. Second, the optical setup, involving the camera viewing angle (θ) and the distance to the object ($d_{object}$), defines the canopy scene to be analyzed, and an appropriate combination of such parameters is needed to reduce the number of nonleaf pixels. Third, since the pixel resolution of thermal cameras is notably low, a single pixel can detect both soil and leaf thermal radiation such that thresholding based solely on temperature can yield a high level of uncertainty in the estimation of $T_{canopy}$. Finally, the studies mentioned above do not provide a detailed explanation regarding the various image processing procedures involved in the use of thermal images; this is possibly due to the dependence on commercial software.

This research has the following specific objectives: (1) To provide a thorough description of infrared thermography for the estimation of canopy temperature, and subsequently the CWSI, in the context of potato production; (2) to generate an open-source software to perform such tasks that can be freely used by the agricultural remote sensing community; (3) to demonstrate that thermographic sensors along with the described acquisition methodology are appropriate for defining irrigation thresholds, which can reduce water consumption.

## 2. Materials and Methods

### 2.1. Plant Material and Study Area

The potato cultivar used was UNICA (CIP code: 392797.22), an early variety adapted to warm and dry environments and slightly tolerant to salinity [24]. Two field trials were performed at the International Potato Center (CIP) and the National Agrarian University—La Molina (UNALM) experimental stations in Lima, Peru (12.08° S, 76.95° W, 244 m.a.s.l.) during October 2017–January 2018 (first experiment—CIP, E1) and June–September, 2018 (second experiment—UNALM, E2). The study site is characterized by a semi-warm and humid climate [25].

During the growing season, the minimum temperature ($T_{min}$), maximum temperature ($T_{max}$), relative humidity (RH), solar radiation ($R_{s}$), and maximum vapor pressure deficit ($VPD_{max}$, estimated using the equation used by Ramírez et al. [26]) were 16.4 ± 0.19 °C, 23.2 ± 0.23 °C, 86.5 ± 0.45%, 16.0 ± 0.43 MJ m^−2^ day^−1^, and 1.06 ± 0.03 kPa, respectively (in E1); and 14.2 ± 0.05 °C, 18.7 ± 0.19 °C, 89.7 ± 0.04%, 7.5 ± 0.46 MJ m^−2^ day^−1^, and 0.64 ± 0.03 kPa, respectively (in E2). Specific values for environmental conditions per experimental period are provided in Table 1 and Table 2.

### 2.2. Experimental Design and Crop Management

The experimental units (EU) were plots of 3.6 × 12.5 m^2^ (with 120 plants) and 4.5 × 15.8 m^2^ (with 180 plants) in E1 and E2, respectively. Since the experiments were conducted in two facilities with different total areas but identical weather conditions, the plot sizes are different. However, to overcome any variability in the results, we adopted the same population density of 3.7 plants m^−2^ per plot in both cases. A completely randomized design, with three irrigation timing treatments repeated in five EUs, was established in both experiments. Two water restriction levels defined with thresholds values of CWSI (T2: CWSI < 0.4 and T3: CWSI < 0.7) were compared against a control (T1: fully irrigated). The reader is referred to Section 2.4 for a detailed description of the CWSI. The water restriction started when the tuber initiation occurred (33 and 31 days after planting (DAP) in E1 and E2, respectively). In both experiments, the drip irrigation system was comprised of a nonpressure compensating emitter and distributed drip-tapes (Toro Aqua-Traxx PBX). There were two drip-tapes per furrow separated by 0.2 m. In addition, the emitter flow rate (the distance between emitters was 0.20 m) was 1.27 L h^−1^ at 0.055 MPa, and the manufacturing coefficient of the variation of the emitters was below 3%. Irrigation pulses (up to field capacity) were supplied to maintain the soil moisture above 70% of field capacity (in T1), and each time the crop reached the established threshold value of CWSI (0.4 and 0.7 in T2 and T3, respectively).

The fertilization application (N:P_2_O_5_:K_2_O) consisted of 180:100:160 and 160:80:180 kg ha^−1^ supplied as NH_4_NO_3_:(NH_4_)_2_HPO_4_:K_2_SO_4_ and NH_4_NO_3_:H_3_PO_4_:K(NO_3_)_2_ in E1 (at sowing and hilling) and E2 (in four periods from emergency to tuber initiation), respectively. Additionally, doses of CaO:MgO (60:30 kg ha^−1^) were supplied as Ca(NO_3_)_2_:Mg(NO_3_)_2_·6H_2_O in E2. Chromatic and pheromone traps were used as the ethological control in both experiments. Recommended doses of chemical products with different mechanisms of action were applied in E1: 0.50 L ha^−1^ of Movento 150 OD (Bayer AG, Leverkusen, Germany), 0.55 L ha^−1^ of Sorba 50 EC (Farmagro, Lima, Peru), 0.60 kg ha^−1^ of Evisetc-S (Arysta Life Science, PA, USA), and 0.15 kg ha^−1^ of Trigard 75 WP (Farmagro, Lima, Peru). These were rotated weekly from 30 to 75 DAP. In E2, a single chemical application composed of 0.40 L ha^−1^ of Confidor 350 SC (Bayer AG, Leverkusen, Germany) was performed at 66 DAP.

### 2.3. Image Acquisition and Analysis

The thermal camera used in the experiments was the FLIR E60 (FLIR Systems Inc., Täby, Sweden), which is a specialized device for hand-held acquisition due to its robust and light-weight design. It includes both IRT and RGB sensors. The IRT camera lens has an angular field-of-view (FOV) of 25° × 19° and focal length (*f*) of 18 mm. The IRT sensor has a spatial resolution of 320 × 240 pixels, a thermal sensitivity of <0.05 °C in the 7.5–13 μm spectral range, a thermal response time of ∼8–12 ms, and an accuracy of ±2 °C or ±2% when reading for an ambient temperature from 10 to 35 °C. The RGB camera lens has an angular FOV of 53°× 41°, and the RGB sensor has a spatial resolution of 2048 × 1536 pixels.

Each pair of images was manually acquired with an object distance $d_{object}$ of ≈3 m (defined as the distance between the camera’s lens and the plant canopy), a horizontal inclination or viewing angle $\theta \;\approx \;60^{{}^{\circ}}$, with respect to the zenith and in position opposite to the sun (see Figure 1A). Viewing angle θ selection relied on the following: (1) The assumption that the potato canopy behaves as a reflective uniform surface wherein higher emission can be acquired when θ is similar to the angle of incidence of solar rays ($\theta_{i}$); (2) the fact that the most suitable period for water status characterization is around 14:00 [8]; and (3) the need for having 75% of the potato canopy in the scene when combined with IRT’s FOV and $d_{object}$. Finally, the acquisition was conducted between 13:00 and 15:00 ($\theta_{i}\in {\left[60,80\right]}^{{}^{\circ}}$) with a constant θ and during clear-sky days to meet the previous requirements and ensure reproducibility. As a result, six plants (effective area ≈ 1 m^2^) were captured in the IRT scene. The acquisition period was previously determined in [8], in which leaf temperature and stomatal conductance were monitored during the entire day (from 07:00 to 18:00) for the same variety of potato (UNICA), comparing well-irrigated and water-stressed potatoes. It was also demonstrated that this variety showed a strong stomatal sensitivity closure after noon due to abrupt changes in the meteorological variables: VPD and solar radiation.

The total thermal radiation received at the detector includes both the radiation emitted by the object ($T_{object}$) and the radiation originating in the surroundings, which is reflected by the object ($T_{reflected}$) [27]. Therefore, the estimated temperature from the IRT cameras needs to be corrected to obtain an accurate value for $T_{object}$. The correction procedure, which is explained below, relies on the a priori estimation of $T_{reflected}$. For this, an IRT image from a low-emissivity cardboard panel with a uniform surface was acquired before each acquisition campaign and the value of $T_{reflected}$ was estimated by strictly following the provided instructions in the FLIR E60 user manual. Additionally, the calculation of CWSI (explained in Section 2.4) requires the measurement of the temperature of a wet leaf reference ($T_{wet}$) [28]. As recommended in [6,21,29,30], a 1 mm double piece white cotton cloth around a piece of polystyrene foam floating in a 0.32 × 0.22× 0.10 m^3^ plastic tray was used as an artificial wet reference surface (AWRS) (Figure 1C). It was present in every IRT image acquired in this study (see Figure 1B).

The proposed methodology to align the RGB and IRT images, extract the canopy area, and calculate the average temperature of the potato canopy in the scene is depicted in Figure 2. For this, let the RGB visible image be defined as $I_{RGB}\in Z_{+}^{W_{RGB}\times H_{RGB}\times 3}$ and the IRT image as $I_{IRT}\in R_{+}^{W_{IRT}\times H_{IRT}}$, where $W_{RGB}\times H_{RGB}$ = 2048×1536 pixels and $W_{IRT}\times H_{IRT}$ = 320×240 pixels are their resolutions, respectively.

Geometric Transformation: The IRT and RGB images acquired with the FLIR E60 camera have a horizontal and vertical displacement vector $\left(b_{x},b_{y}\right)$ (units in pixels) due to the shift between both image sensor’s optics. In addition, there is an additional scaling factor $\left(S_{x},S_{y}\right)$ (unitless) as a result of the different lenses’ FOV and the corresponding sensors’ spatial resolution (pixel quantity and size). Such parameters can be calculated a priori using a one-time calibration protocol based on the methodology described in [13]. Fifty pair of IRT and RGB images acquired with $\theta \;\approx \;60^{{}^{\circ}}$ and $d_{object}\;\approx \;3$ m over the potato crop, and including the AWRS in the scene, were utilized for calibration. A MATLAB script was developed to allow the user to manually select correlated control points in both images through visual inspection, and consequently, provide averaged displacement vector and scaling factors (Figure 3). The presence of the AWRS in both images serves to locate additional control points due to its higher contrast (see Figure 3). The number of image pairs (50) helps overcome the inherent human error in the selection, and thus, reduce the measurement uncertainty. The resulting scaling factor $\left(S_{x},S_{y}\right)=\left(0.367,0.375\right)\pm 5.40\%$ is firstly applied to the RGB image ($I_{RGB}$). Bi-linear interpolation in $I_{RGB}$ is allowed since it only serves to determine the canopy location in the scene. Then, the displacement vector $\left(b_{x},b_{y}\right)=\left(219,189\right)\pm 2.33\%$ is applied to the IRT image ($I_{IRT}$) with respect to $I_{RGB}$. The coordinate origin (0,0) is assumed to be the top left corner of $I_{RGB}$. Furthermore, no scaling was applied, i.e., the interpolation process is carried out over $I_{IRT}$ to avoid altering the IRT sensor measurements. Figure 2 (Step 1) shows the resulting scaled $I_{RGB}$ and translated $I_{IRT}$ as a single false-color image, which also indicates the overlapping region to be analyzed.Color-Based Threshold Calculation: The green–red vegetation index (GRVI) is used to determine the presence of the potato canopy in the scene. As was thoroughly explained and assessed in [31], GRVI can serve as a threshold to determine the leaves’ location in an RGB image. After the geometric transformation procedure is performed, the overlapped region from the scaled RGB image is extracted; it is referred as $I_{RGB}^{*}$ from now on. Then, the GRVI is calculated for each pixel (h,w) using the following equation:
(1)$$GRVI=\frac{I_{RGB}^{*}\left(h,w,2\right)-I_{RGB}^{*}\left(h,w,1\right)}{I_{RGB}^{*}\left(h,w,2\right)+I_{RGB}^{*}\left(h,w,1\right)},$$
where $I_{RGB}^{*}\left(h,w,2\right)$ indicates the pixel located at the 2D coordinate (h,w) in the green layer (red=1, green=2 and blue=3). In [31], a value of GRVI = 0 can detect the early phase of leaf green-up over several forest species, such as deciduous broadleaf and deciduous coniferous, as well as grassland and a rice paddy field. In this study, additional experiments were conducted with the potato canopy, and it was found that a threshold value of GRVI=0.04 can determine the presence of the potato canopy with a high level of accuracy when it is not affected by the crop senescence. Finally, the logical mask $M\in {\left\{0,1\right\}}^{320\times 240}$ is generated and contains 1s when a given pixel yields a GRVI≥0.04 and 0 s otherwise.Morphological Operations: As a result of the linear interpolation of the RGB image $I_{RGB}$, individual pixels do not conserve their spectral information, and the GRVI cannot detect them as part of the canopy, ultimately creating small holes (a group of 0s) in the mask *M*. For this reason, a set of mathematical morphological operations is used. First, dilation is applied to eliminate the noiselike structures over the potato canopy. Second, erosion is used to fully cover those regions that do not belong to the canopy, as shown in see Figure 2 (Step 3). Only those values with 1 s (white in the image) are used to calculated the averaged $T_{canopy}$. Additionally, since the size of the image is fixed (320 × 240 pixels), a kernel size of 4 × 4 pixels was used for both operations. Finally, small regions with 1/10 of the total mask area are removed.Correction with the FLIR Metadata and Average Temperature Calculation: The IRT image $I_{IRT}$ provided by FLIR E60 has units of Kelvin and its generation considers the total IR radiation that reached the detector during acquisition. Such radiation is composed of two components: the thermal radiation originated from the object and the radiation originating in the surroundings and reflected by the object. The fraction of the reflected radiation depends on the emissivity of the object ε, specifically, when ε<1, and should be removed from the measurement [7]. In order to perform the correction, the FLIR E60 provides the estimated total temperature in raw 16-bit format *S* and additional factory calibration parameters. Additionally, $T_{reflected}$ and ε=0.96 [6,8] are utilized to estimate $T_{object}$ as follows:
(2)$$RAW_{reflected}=\frac{R_{1}}{R_{2}\left(exp\left(B/T_{reflected}\right)-F\right)}-O,$$
(3)$$RAW_{object}=\frac{S-\left(1-\varepsilon \right)RAW_{reflected}}{\varepsilon },$$
(4)$$T_{object}=\frac{B}{\ln\left(R_{1}/\left(R_{2}\left(RAW_{object}+O\right)\right)+F\right)},$$
where $R_{1}$, $R_{2}$, *F*, *O*, and *B* are the so-called Planck calibration constants for FLIR cameras (and are set upon manufacturing), which can be extracted by the publicly-available EXIFTOOL application from the IRT image metadata. Furthermore, $T_{reflected}$ is the reflected temperature in Kelvin, $RAW_{reflected}$ is a 16-bit calculated value, *S* is the 16-bit raw value provided by the FLIR R60 camera, and $T_{object}$ is the object temperature in Kelvin. This procedure is performed for each pixel in the IRT image. Such values are then multiplied with the logical mask *M* and averaged to obtain the average canopy temperature $T_{canopy}$. It is noteworthy that the noise likely induced by manual acquisition is highly reduced through the averaging of multiple pixels in five (5) IRT canopy images for a given treatment.

The selected emissivity value ϵ=0.96 was thoroughly studied in [6,8,32], and in this study, it was considered constant throughout the complete growing season. As pointed out by Usamentiaga et al. [33], the emissivity of a real object is variable and dependent on wavelength; however, it can be assumed constant when the wavelength interval is short. Since the IRT image spectral width is 6 μm, the assumption of it being constant ϵ remains valid. Finally, the described algorithm has been implemented as a software package named the “Thermal Image Processor” (TIPCIP) using QT Creator v5.0 for Windows operating system and under an open-source and free-access policy. The download links for the executable and source files are included in the Reference section.

### 2.4. Response Variables

In E1, six thermocouples (TT-T-36-SLE, Omega Engineering Inc., Manchester, UK) with an accuracy greater than 0.5 °C or 0.4% (above 0 °C) were attached to each plot into the abaxial surface of the leaf center of target plants using surgical tape [34]. In both experiments, thermal images were acquired (interdaily from 34 to 82 DAP) and processed according to the methodology described in Section 2.3. Data from thermocouples and IRT images correspond to the same plots. Canopy temperature data from IRT images were used to estimate the CWSI following the empirical method used for potatoes in [12]:(5)$$CWSI=\frac{T_{canopy}-T_{wet}}{T_{dry}-T_{wet}},$$
where $T_{canopy}$ is the measured canopy temperature, $T_{wet}$ is the AWRS measured temperature, and $T_{dry}$ is the artificial dry reference surface estimated temperature. $T_{dry}$ was considered 13 °C [8] and 7 °C [6,12] over the air temperature in E1 (hot season) and E2 (wet season), respectively.

Before the irrigation was applied in each treatment, soil samples were collected at 0–0.35 m depth (where >80% of root-zone biomass is concentrated) in each plot to estimate volumetric water content ($\theta_{v}$, in %) according to Ramírez et al. [6]. The irrigation time and irrigated water quantity (IWQ, in mm) was estimated in each treatment from $\theta_{v}$, $\theta_{v}$ at field capacity (32.7% and 28.4% in E1 and E2, respectively), root-zone width (0.40 m), root-zone depth (0–0.35 m), and drip-tape flow rate (8.5 and 7.7 L m^−1^ h^−1^ at 0.05 MPa in E1 and E2, respectively) (see the equations in [6]). Before treatment onset, five intense irrigation pulses were realized by furrow irrigation in E1 (in all treatments) at 1, 8, 12, 19, and 25 DAP to every plot with ≈ 36 mm per irrigation. In this case, the irrigated water volume was estimated as the product of the flow of the main water channel of CIP’s experimental station (≈0.75 m^3^ min^−1^) and the time required to flood each plot. In the case of E2, the drip-tape irrigation system was utilized as this facility (UNALM) did not have a water channel for crop flooding. Thus, the irrigation (in all treatments) before tuber initiation (31 DAP) occurred when soil moisture achieved 70% of field capacity. A uniform-distributed irrigation utilized 1131.6 m^3^ ha^−1^ of water.

Total IWQ (IWQ_*T*_) is the sum of IWQ (each irrigation) during the growing season. Four center plants (in which the CWSI evaluations were performed) were individually harvested and separated into leaves, stems, and tubers at 102 and 92 DAP in E1 and E2, respectively. Fresh tuber yield (FTY, in t ha^−1^) was calculated from the average value of tubers biomass per plant and plant density.

### 2.5. Statistical Analysis

The effects of irrigation treatments on CWSI for each assessment were tested with one-way ANOVA. Fisher’s least significant difference (LSD) test was performed to determine differences among irrigation treatments on FTY and IWQ_*T*_. Pearson correlation coefficient ($r_{Pearson}$) was calculated to determine the accuracy of leaf temperature estimations. The significance of all statistical tests were assessed at *p* < 0.05 and *p* < 0.01 using R v3.6.1 software [35].

## 3. Results

### 3.1. Accuracy of the Canopy Temperature Estimations

The range of values of $T_{canopy}$ recorded by the thermocouples was between 23.0 and 39.6 °C. The overall average values of $T_{canopy}$ estimated by the TIPCIP software showed a positive linear relationship with the average $T_{canopy}$ values by thermocouples ($r_{Pearson}$ = 0.84, *p*-value < 0.01) (Figure 4). In addition, 87.0% and 82.6% of the $T_{canopy}$ values using TIPCIP, corresponding to the T1 and T2 treatments, underestimated $T_{canopy}$ as compared to the thermocouple values, in a range from −2.05 to −0.15 and −3.27 to −0.61 °C, respectively (Figure 4). Furthermore, 60.9% of T3 treatment values overestimated $T_{canopy}$ as compared to the thermocouple values, in a range of +0.21 to +3.80 °C (Figure 4). The accuracy of $T_{canopy}$ estimations was better in T1 ($r_{Pearson}$ = 0.90, *p*-value < 0.01), followed by T2 ($r_{Pearson}$ = 0.85, *p*-value < 0.01), and T3 ($r_{Pearson}$ = 0.83, *p*-value < 0.01).

### 3.2. CWSI and Irrigation Treatments through the Growing Period

During E1, 14, 9, and 1 irrigation treatments of between 4 and 14 mm, 5 and 16 mm, and 25 mm (Figure 5A) were established for T1, T2, and T3, respectively. The maximum average value of IWQ_*T*_ and FTY in this trial were 3238.7 ± 95.80 m^3^ ha^−1^ and 38.9 ± 5.94 t ha^−1^ corresponding to T1 (Figure 6A,B) with no significant (*p*-value > 0.05) reduction in FTY with T2 of 34.5 ± 3.72 t ha^−1^, which showed a −10.5% decrease in IWQ_*T*_. T3 significantly (*p*-value < 0.05) reduced by −37.2% and −61.4% the IWQ_*T*_ and FTY, respectively, in comparison to T1 (Figure 6A,B). During E2, 17, 10, and 4 irrigation treatements of between 1.7 and 9.2 mm, 3.7 and 18.7 mm, and 3.7 and 16.5 mm (Figure 5B) were established for T1, T2, and T3, respectively. The maximum average value of IWQ_*T*_ and FTY in this trial was 1738.5 ± 44.35 m^3^ ha^−1^ and 49.5 ± 3.37 t ha^−1^ achieved by T1. T2 showed a no significant (*p*-value > 0.05) reduction in FTY in comparison to T1 of 44.3 ± 2.66 t ha^−1^, but showed a significant −29.7% (*p*-value < 0.05) reduction in IWQ_*T*_ (Figure 6C,D). T3 significantly (*p*-value < 0.05) reduced by −55.0% and −39.2% in terms of the IWQ_*T*_ and FTY, respectively, in comparison to T1 (Figure 6C,D).

The CWSI values ranged between 0.25 and 0.64, 0.31 and 0.60, and 0.39 and 0.82 (in the hot season—E1; Figure 5A), and 0.14 and 0.59, 0.12 and 0.73, and 0.12 and 1.0 (in the wet season—E2; Figure 5B) in T1, T2, and T3 respectively. The average value of CWSI before the irrigation treatments in T1, T2, and T3 were 0.34 ± 0.02, 0.45 ± 0.02, and 0.72 ± 0.0 (in E1); and 0.34 ± 0.04, 0.56 ± 0.05, and 0.65 ± 0.03 (in E2), respectively. In E1, the CWSI values were close to the established threshold (0.4 and 0.7 for the T2 and T3 treatments, respectively). In E2, the CWSI values exceeded the threshold after 60 DAP in the T2 treatments (five of nine measurements; Figure 5B) and T3 (two of nine measurements; Figure 5B).

## 4. Discussion

### 4.1. Acquisition Configuration and Comparison with Instrumental Measurements

With the advent of relative inexpensive thermographic imager systems such as those that include both IRT and RGB sensors, the analysis of larger crop areas can be attained with a single pair of images. The additional RGB sensor provides the capability to determine specific structures in the scene through appropriate image processing techniques. Notwithstanding, several factors have to be considered when working with these images, such as the field-of-view of the optical system, the distance to the object, and the viewing angle, as they could insert uncertainty in the temperature estimation.

The selection of an adequate viewing angle requires the analysis of the three-dimensional structure of the canopy. For a highly heterogeneous canopy, the best viewing angle is parallel to the solar beam line-of-sight (LOS) since the image will mostly include well-illuminated leaves and reduce the number of shaded, cooler leaves that ultimately would affect the estimation of $T_{canopy}$ [7]. On the contrary, a potato canopy, during 30 to 80 DAP, can be assumed to be an almost planar surface where leaves have quasi-uniform heights from the floor. The selected viewing angle of 30° inclination from the horizon not only allows one to acquire this uniform structure in a single image with a reduced number of shaded leaves, but also prevents the presence of shadows in the scene, which can appear if the image is taken parallel to the solar LOS. In addition, since the fixed camera’s FOV is narrow, substantial variations in the reflected emission from different parts of the generated image are not expected. However, the assumption of a planar canopy is not valid for the days before 30 DAP as a result of the growth stage, nor for days after 80 DAP as a result of the senescence period. It is noteworthy that Luquet et al. [36] investigated the impact of the viewing angle on the estimated temperature of different canopies, presenting variations of up to 1.5 °C for viewing angles higher than 45° with respect to the ground plane normal and opposite to the sun. According to [37], such reported temperature variations could be governed by the changing proportion of background (soil in the scene) viewed at different angles. As this is still an open discussion, further analysis should be done for plant canopies whose 3D structure significantly deviates from planar.

The spatial resolution (i.e., the size of a resulting pixel in the image typically in units of centimeters or meters) varies with the camera–object distance $d_{object}$. Short distances yield a more extensive area of coverage for a pixel, which could include both canopy and soil, thereby hindering the correct temperature estimation. This issue is aggravated when a nadir view is utilized to acquire the images from a sparse crop plot [37]. Subpixel methods, which can estimate the fraction of the pixel only occupied by the leaf, have been thoroughly studied in [38]; however, the high complexity of the algorithm and the computational cost of its implementation limit its utility for campaigns that can span several months. In this study, the use of a viewing angle with inclination from the horizon along with a $d_{object}\approx$ 3 m and the given IRT FOV resulted in >≈75% of pixels with canopy. Although the image acquisition procedure was conducted manually, instead of using a fixed platform, a high correlation between thermographic estimated temperatures and in-situ measurements from thermocouples was attained. Such a result can be leveraged by newly introduced thermographic systems, such as the FLIR ONE (FLIR Systems Inc.,Täby, Sweden) in which a thermal camera can be attached to a smartphone. In this context, this research also hopes to make the technology available to smallholder farmers, who can acquire these inexpensive devices, acquire data manually, and use the developed software to, ultimately, conduct simple irrigation management.

Additionally, a comparative study between thermography and in-situ measurements was conducted in this research. It is noteworthy that physical techniques for measuring leaf temperatures are different. Thus, thermocouples utilize thermal conduction through physical contact with the abaxial surface. Transmission of heat from the plant towards the sensor is carried out through the movement of excited electrons from the plant to the sensor, which are ultimately converted into voltage and mapped to a temperature value. On the other hand, infrared thermal cameras measure thermal emissions from the adaxial surface, which is exposed to the convection effect (air circulation) and direct incident light. Through the Planck’s law equation, IR radiation is then converted into temperature values. Although a large temperature gradient between adaxial and abaxial surface due to its millimeter thickness is not expected, the use of different methods imposes additional uncertainty, which is finally revealed in the correlation calculation.

### 4.2. Thermography Usefulness for Irrigation Scheduling in Potato Crops in Humid Environments

Humid environments, characterized by low VPD, impose technical (emission detection, image processing of thermal and visible images; [37]) and physiological (stomatal openness sensitivity detected in the potato; [8]) challenges for plant water status using thermography. The dry temperature in the CWSI calculation represents the temperature that no transpired leaves achieve [28], and from an empirical perspective (sensu [12]), it is calculated adding a *X*-value to the air temperature (Equation (Equation 5)). *X*-value can be experimentally determined as the maximum temperature achieved by leaves previously covered with petroleum jelly to avoid transpiration [28]. For the potato, seven [6,12] and 13 °C [8] have been used as *X*-values in different areas. The potato has an acute stomatal closure sensitiveness in humid areas during the day, depending on various thresholds of VPD and radiation, which promotes an important increase in leaf/canopy temperature in relation to the atmospheric temperature [8]. In this study, different *X*-values were used in the same area depending on the season, which was characterized by different average values of VPD and radiation. Thus, the hot and wet season showed values (99% of the data) of solar radiation of 14.2–17.9 and 3.7–6.2 MJ m^−2^ day^−1^, with average VPD values (during the evaluations) of 0.90 ± 0.01 and 0.39 ± 0.01 kPa; 13 and 7 °C being the *X*-values used for each season, respectively. Because of the close relationship between the “standard” and maximum stomatal conductance at light-saturated and tuber yield, CWSI has been proposed as a good indicator of water status in potato crops [6,8,12,39]. In agreement with other works [6,8], this study confirms the use of CWSI <0.4 as a threshold for irrigation in potato crops, which allowed us to save water (341.6 ± 63.7 and 515.7 ± 37.7 m^3^ ha^−1^ in E1 and E2, respectively) without a significant (*p*-value > 0.05) reduction in tuber yield. Our study area belongs to the South American arid diagonal [40], a hyper-arid zone distributed along the Peruvian coast, characterized by a higher atmospheric humidity during the winter season (May–August) promoted by fogs that come from the Pacific Ocean [41,42,43]. This wet season is the most appropriate for potato production in this zone [44], and the most challenging for the use of thermography methodology (see discussion in Section 4.1). Our second trial was carried out during this season, obtaining the potential tuber yield reported for the UNICA potato variety in the literature (50 t ha^−1^; [24]) with a significant (*p*-value < 0.05) water saving (515.7 m^3^ ha^−1^) in relation to the control using 0.4 CWSI threshold value (T2, Figure 6D). In comparison to the hot season trial, the wet season was monitored three times per week (Mondays, Wednesdays, and Fridays), and in some cases, our target thresholds were mainly surpassed after 60 DAP (Figure 5B) i.e., after maximum canopy cover and during the senescence stage. Ramírez et al. [6] caution to avoid potato plants close or above a CWSI of 0.6 (known as the “severity threshold”) because of the potential of affecting tuber yields through oxidative damage. However, it seems that this severity threshold could be achieved after maximum canopy cover or during senescence in the wet season without affecting the tuber yield. The Peruvian central coast will be likely affected by an increase in temperatures in the future because of global warming [45]. The hot season trial simulated the potato production under these future scenarios, which will demand more water and potential yields in this crop will be difficult to achieve.

### 4.3. Advantages and Disadvantages of the Developed Software (TIPCIP)

Several agricultural analyses based on IRT imagery and image processing did not provide a user-level software; this is because the authors were mainly concerned about validating the physiological meaning of IRT-based treatments [21,23,37]. As a result, similar software packages were implemented but not distributed to the scientific community. In this context, this research not only reveals the various steps involved in canopy identification and temperature calculation using IRT images, but also offers the open-source TIPCIP software to perform those tasks. Hence, TIPCIP allows the user to automatically identify the canopy section in the scene and provides the average temperature, processing up to 50 image pairs (RGB-IRT) in a single run, and reporting formatted (CSV, XLS, among others) results. Furthermore, as a result of the available access to the source code, a scientific developer could include additional image processing steps to adapt the software to other crops or to the use of NDVI images instead of RGB images, thus increasing the accuracy of the canopy detection. Notwithstanding, TIPCIP is currently limited to working with FLIR-format images for the extraction of raw data as well as calibration parameters. However, a developer could include additional libraries or functions facilitating it to read other formats and increase the current structure of the software.

In addition, commercial thermal image processing tools for general purposes can be found on the internet. For example, the FLIR Tool Plus (U.S. $250) enables the user to visualize, edit, and analyze IRT images acquired with FLIR cameras. However, such analyses are conducted only with thermal images, and consequently, image classification/segmentation is solely performed based on temperature thresholds. A zero-cost programming tool, which is still in development, can be found at https://github.com/micasense/imageprocessing for Micasense cameras. The Altum Micasense multispectral camera includes an IRT sensor and allows the users to generate their code based on the Python programming language. The user requires specific programming skills to interpret the examples but then can construct software and include segmentation as well as generate several vegetation indexes.

In sum, TIPCIP provides an option for image processing related to CWSI generation based on FLIR-format images. Its source code allows for modification but requires programming skills in the C++ programming language.

## 5. Conclusions

The accurate estimation of a plant water stress index via thermal imaging amounts to the need for a simple and efficient image processing algorithm. In this study, the Thermal Image Processor (TIPCIP) software was implemented, and through image segmentation and morphological operations, determined the canopy region and its average temperature. A comparison with in-situ measurements using thermocouples showed that our estimated values of $T_{canopy}$ yield were in good agreement ($r_{Pearson}$ = 0.84). On the basis of these results, an irrigation schedule was implemented using a CWSI-based threshold, specifically CWSI < 0.4 and CWSI < 0.7 for T2 and T3, respectively, in two experiments carried out during the hot and wet season. A water saving without yield penalization was found for T2 (341.6 ± 63.7 and 515.7 ± 37.7 m^3^ ha^−1^) in E1 and E2, respectively. Finally, the authors have made the developed TIPCIP software available under the Creative Commons (CC 4.0) policy and encourage the agricultural remote sensing community to assess it and provide feedback (the executable files can be downloaded from [46] and source code files from [47]).

## Figures and Tables

**Figure 1 sensors-20-00472-f001:**
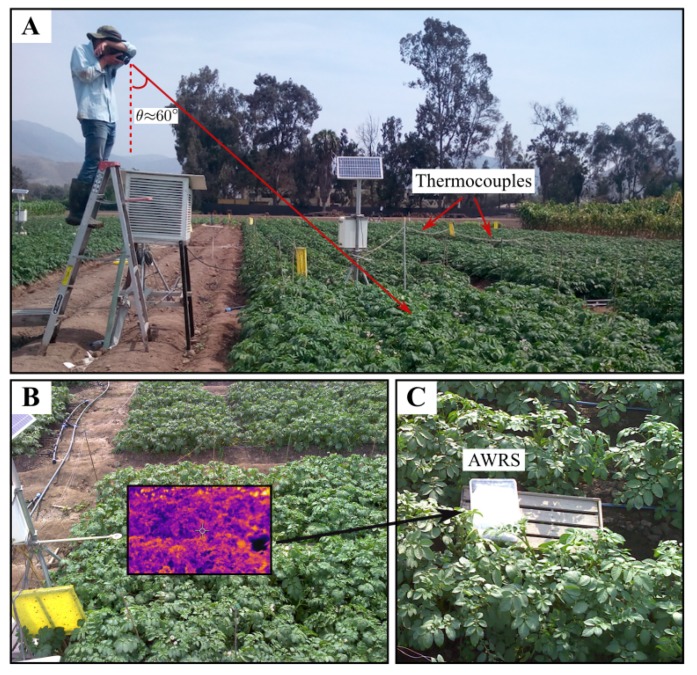
Acquisition procedure of infrared thermography (IRT) and red-green-blue (RGB) images using the FLIR E60 model thermal camera, according to the procedure of Rinza et al. [8] (**A**). The IRT and RGB images acquired in A (**B**). AWRS—the artificial wet reference surface (**C**).

**Figure 2 sensors-20-00472-f002:**
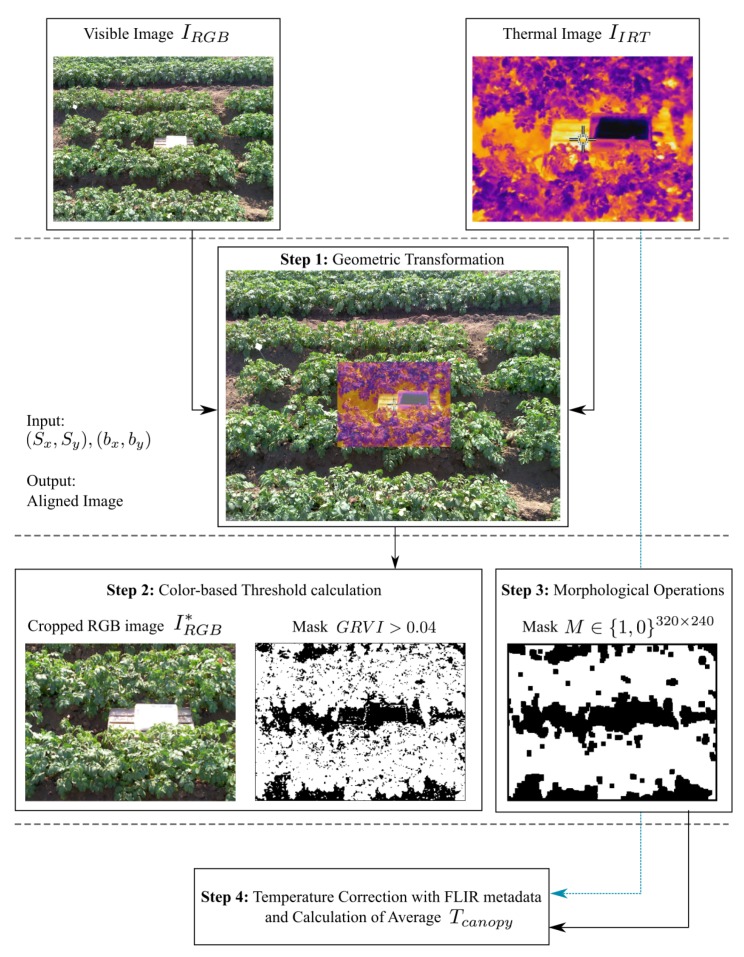
Automatic canopy average temperature algorithm flowchart. The IRT image is presented in a false color pattern for better visualization. Images are not presented in real scale.

**Figure 3 sensors-20-00472-f003:**
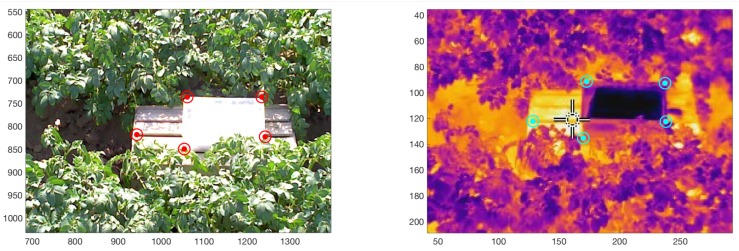
Manually selected, spatially correlated control points in both RGB and IRT images to estimate the geometric transformation parameters: scaling factors ($S_{x},S_{y}$) and displacement vector ($b_{x},b_{y}$).

**Figure 4 sensors-20-00472-f004:**
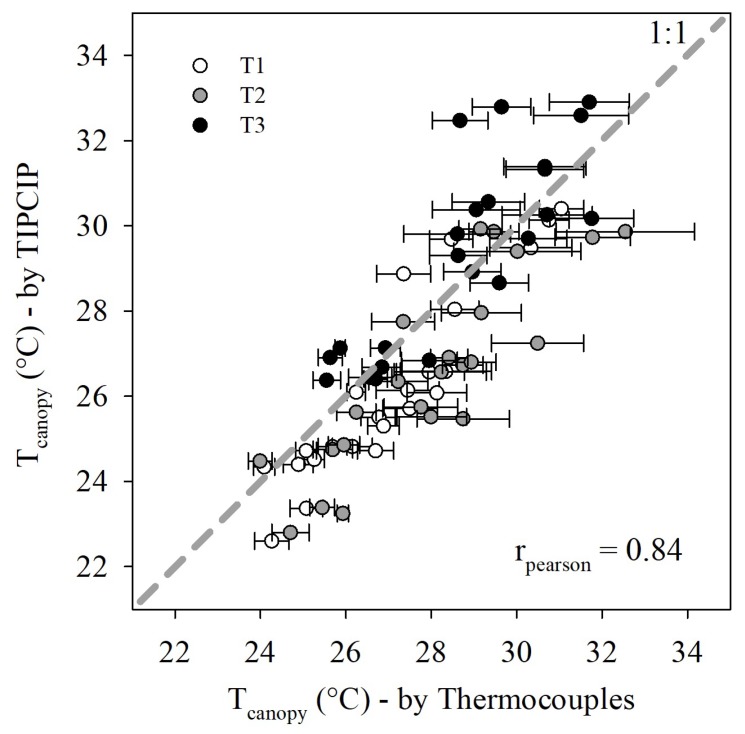
Scatter plot (±standard error) of canopy temperatures ($T_{canopy}$) acquired by thermocouples (*x*-axis) and infrared thermal images, which were processed with the Thermal Images Processor (TIPCIP) software (*y*-axis) under three irrigation treatments: T1 (control or fully irrigated), T2 (crop water stress index (CWSI) < 0.4), and T3 (CWSI < 0.7). The 1:1 dashed line (*x* = *y*) is plotted as a reference. These data were collected in the first experiment (E1). $r_{Pearson}$: Pearson correlation coefficient.

**Figure 5 sensors-20-00472-f005:**
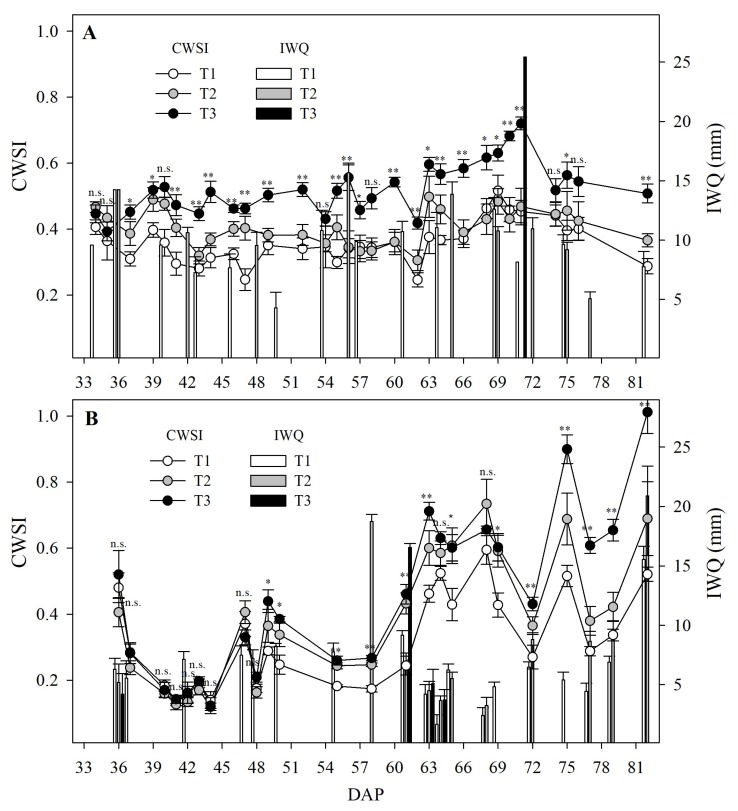
Average value (± standard error) of crop water stress index (CWSI, line chart) measured before irrigation treatments, and irrigated water quantity (IWQ, bar chart) for the first (**A**) and second (**B**) experiment for the three irrigation treatments: T1 (control or full irrigation), T2 (CWSI < 0.4), and T3 (CWSI < 0.7). The terms **, *, and n.s. indicate p<0.01, p<0.05, and p>0.05 (not significant), respectively, in the ANOVA. DAP—Days after planting.

**Figure 6 sensors-20-00472-f006:**
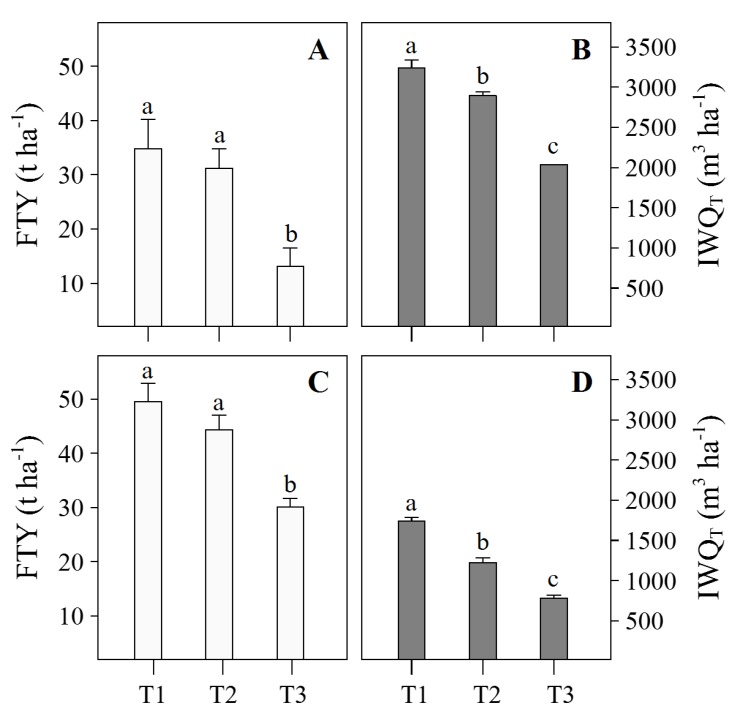
Average value (±standard error) of fresh tuber yield (FTY) and total irrigated water quantity (IWQ_*T*_) in the first (**A**,**B**) and second (**C**,**D**) experiment. The different letters in each subfigure indicate significant differences (*p* < 0.05) between treatments (T1: Control or full irrigation, T2: CWSI < 0.4, and T3: CWSI < 0.7) detected by LSD test. CWSI—Crop water stress index.

**Table 1 sensors-20-00472-t001:** Environmental conditions during the experimental period 2017–2018 (E1). Average daily values ± standard error. VPD—Vapor pressure deficit.

	October	November	December	January
Minimum Temperature (°C)	14.6 ± 0.08	15.4 ± 0.19	17.5 ± 0.18	19.7 ± 0.10
Maximum Temperature (°C)	21.5 ± 0.27	22.2 ± 0.26	24.1 ± 0.32	27.5 ± 0.21
Average Temperature (°C)	16.8 ± 0.13	17.8 ± 0.18	20.0 ± 0.22	22.7 ± 0.13
Relative Humidity (%)	83.4 ± 0.49	80.6 ± 0.54	81.0 ± 0.73	74.4 ± 0.66
Solar Radiation (MJ m^−2^ day^−1^)	17.2 ± 0.63	15.9 ± 0.73	14.4 ± 0.95	18.5 ± 0.65
Average VPD (kPa)	0.35 ± 0.01	0.42 ± 0.01	0.48 ± 0.03	0.76 ± 0.03
Maximum VPD (kPa)	0.84 ± 0.03	0.92 ± 0.03	0.99 ± 0.05	1.51 ± 0.05

**Table 2 sensors-20-00472-t002:** Environmental conditions during the experimental period 2018 (E2). Average daily values ± standard error. VPD—Vapor pressure deficit.

	June	July	August	September
Minimum Temperature (°C)	14.8 ± 0.09	14.4 ± 0.06	13.9 ± 0.07	14.3 ± 0.08
Maximum Temperature (°C)	17.8 ± 0.28	18.6 ± 0.37	18.9 ± 0.27	20.4 ± 0.28
Average Temperature (°C)	15.8 ± 0.09	15.7 ± 0.10	15.5 ± 0.10	16.3 ± 0.11
Relative Humidity (%)	85.9 ± 0.56	84.9 ± 0.61	82.5 ± 0.64	80.2 ± 0.59
Solar Radiation (MJ m^−2^ day^−1^)	3.2 ± 0.38	5.2 ± 0.53	7.5 ± 0.66	10.7 ± 0.68
Average VPD (kPa)	0.26 ± 0.01	0.28 ± 0.01	0.32 ± 0.01	0.39 ± 0.01
Maximum VPD (kPa)	0.47 ± 0.03	0.58 ± 0.05	0.68 ± 0.04	0.85 ± 0.04

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
