# Peer review of "Development of an Open-Source Thermal Image Processing Software for Improving Irrigation Management in Potato Crops (*Solanum tuberosum* L.)"

_sensors, 2020, doi:10.3390/s20020472_

Round 1
Reviewer 1 Report
The paper presents an open-source software, that includes image processing techniques, to estimate the crop water stress index in plants.
It is an interesting topic within the scope of the Journal.
In terms of the format, the paper is well organized and written. The scope is clear and well presented. The text is simple, clear and English sounds good, but the assessment by a native person will be better. As for content, it is clear about its contribution and value concerning the other works. However, the paper has some issues which should be better justified, and I would recommend improving your paper clarifying/emphasizing the following topics:
- The literature review could be better, as there are several more recent papers about IRT on agriculture, such as e.g.:
Contribution to Trees Health Assessment Using Infrared Thermography; DOI: 10.3390/agriculture9080171
Infrared Thermography Applied to Tree Health Assessment: A Review ; DOI: 10.3390/agriculture9070156
- I am not at all convinced about the method of recording thermograms in this context (research study), namely without attaching the camera to a tripod so as to adjust the viewing angle and to prevent the natural shaking of the hands. Please clarify.
-Following the question above, it is not clear whether the work is of qualitative or quantitative thermography. Considering the analysis, it seems to be from QIRT, but if so there are several missing details that should be addressed. Please clarify.
- Nothing is said about determining emissivity. This parameter is relevant. I think the authors used 0.96, but no plausible justification is mentioned. In fact, no justification or attempted measurement is given.
-Relevant data, such as air temperature and relative humidity during the experiments, are not reported;
- If the measurement is done during the day, the observation and temperatures obtained are severely influenced by solar radiation and depend largely on the angle of observation, so it seems to us that these measurements should take place after sunset see p .ex., DOI: 10.3390/agriculture9080171
- The accuracy of the auxiliary thermometry system (thermocouples), which we believe to be T-type, is not specifically presented. Once an error analysis is made, it is strange not to specify this parameter ...on the other hand, not knowing this parameter, and the accuracy of the IRT camera, how do you consistently made the analysis and define the significant digits? ...
Most of the above ideas are important to describe the system characteristics and the calibrating procedure in order to reinforce your work.
These interesting references are also recommended to add to the paper:
Al-doski, J.; Mansor, S.B.; Shafri, H.Z.B.M. Thermal Imaging For Pests Detecting-A Review. Int. J. Agric. For. Plant. 2016, 2, 10–30.
Goh, C.L.; Abdul Rahim, R.; Fazalul Rahiman, M.H.; Mohamad Talib, M.T.; Tee, Z.C. Sensing wood decay in standing trees: A review. Sens. Actuators A Phys. 2018, 269, 276–282.
Sharma, S.; Kaushik, A. Views of Irish Farmers on Smart Farming Technologies: An Observational Study.AgriEngineering 2019, 1, 164–187
Infrared Thermography for Temperature Measurement and Non-Destructive Testing, Sensors 2014, 14, 12305-12348; doi:10.3390/s140712305
I hope these few questions/suggestions will help improve your paper.
Accept my best regards.
Reviewer 2 Report
This paper presents a procedure using thermal images for potato irrigation scheduling. Promising results were obtained in a field experiment that was replicated under two different conditions. However, some details of experimental procedure need further justification.
Specific comments and suggestions:
Line 17. I would suggest using as keywords different words than those that appear in the title of the article. Lines 78-88. Instead of summarizing the paper, I would suggest clearly highlight the objectives of the present work. Lines 96-99. I would suggest removing average values since the important values here were those observed during each experiment, which are shown on lines 99-103. Lines 105-106. Why the experimental units were different in the two experiments carried out? A brief explanation is missed. Line 112. Some additional details of the drip irrigation systems are missed: type of emitters (pressure compensating or not), emitter discharge, manufacturing coefficient of variation of the emitters, distance between emitters and distance between driplines. Line 184. Change to “developed to allow”. Lines 234-235. Values for emitter flow rate (L m-1 h-1) seem more of dripline flow rate (since they are related lo length) than for emitter discharge (L h-1). Line 236. Why the five irrigation events by furrow irrigation were only carried out in the first experiment? The range of the irrigation water amount in these first furrow irrigation events would be an interesting information. Line 253. Besides the correlation coefficient, significance level (P-value) of the correlation should be provided. Figure 4. I would suggest showing the meaning of white, grey and black circles in a legend within the figure, but not in the figure caption. Lines 268-270, 275, and 349. Please, provide the significance level (P-value) considered when significant differences are commented. Figure 5. I am wondering if T3 was under control in the first experiment since an irrigation event the highest amount was needed on day 71 in the first experiment. Stress indexes were also the maximum for this treatment at the end of the second experiment. Figure 5. I would suggest showing the meaning of white, grey and black colors in a legend within a figure. In my opinion, this would be help the readers. Discussion is limited to the comparison of thermographic devices and the performance of these sensors for managing irrigation events. However, since the tittle of the paper refers to a software, I would suggest discussing some issues of the developed software such as its advantages/disadvantages regarding other available software. Lines 372-373. In my opinion, conclusions should be more specific. Thus, I would recommend showing clearly the thresholds considered in this study and provide the amount of water saved.
Reviewer 3 Report
It is a very well organized paper, worthy for publication.
